# Metabolic Features of Saliva Before and After Breast Cancer Surgery

**DOI:** 10.3390/metabo15110693

**Published:** 2025-10-25

**Authors:** Denis V. Solomatin, Elena A. Sarf, Lyudmila V. Bel’skaya

**Affiliations:** 1Department of Mathematics and Mathematics Teaching Methods, Omsk State Pedagogical University, 644099 Omsk, Russia; solomatin_dv@omgpu.ru; 2Biochemistry Research Laboratory, Omsk State Pedagogical University, 644099 Omsk, Russia; sarf_ea@omgpu.ru

**Keywords:** saliva, breast cancer, metabolites, mastectomy, surgery, enzymes, phenotype

## Abstract

**Background:** Changes in salivary metabolites in patients after surgery can provide important information for fundamental research. **Objectives**: To analyze changes in the salivary metabolic profile before and after breast tumor surgery. **Methods**: The study involved 660 breast cancer patients (54.6 ± 1.9 years) and 127 healthy volunteers (49.3 ± 1.5 years). Saliva samples were collected from all patients before treatment, and levels of total protein, urea, α-amino acids, imidazole compounds, and nitric oxide, as well as gamma-glutamyl transferase (GGT), lactate dehydrogenase (LDH), α-amylase, and catalase activity, were determined. In 139 breast cancer patients, these parameters were re-measured in saliva 4 weeks after surgical removal of the tumor (radical mastectomy). **Results**: In breast cancer, the activity of GGT (+17.6%, *p* < 0.0001), catalase (+14.7%, *p* = 0.0485), urea content (+24.6%, *p* = 0.0006), total protein (+13.6%, *p* = 0.0432), α-amino acids (+3.1%, *p* < 0.0001) increased in saliva, but LDH activity (−16.9%, *p* < 0.0001) and the content of imidazole compounds (−25.2%, *p* < 0.0001) decreased. It was found that after surgical treatment of breast cancer, a number of the biochemical parameters of saliva are restored. It is shown that the greatest deviations of the metabolic profile of saliva from the norm before surgery correspond to the early stages and the most prognostically favorable phenotypes of breast cancer. After surgery, metabolic differences are most pronounced for triple-negative breast cancer. **Conclusions**: A time interval of 4 weeks is not sufficient for complete recovery, but for most biochemical parameters the values are intermediate between those for healthy controls and primary breast cancer.

## 1. Introduction

Breast cancer is the leading oncological pathology in women aged 30 to 70 years [1,2]. The incidence of breast cancer continues to grow, increasing by 1% per year [3]. A distinctive feature of breast cancer is its heterogeneity, which causes wide variations in prognosis and treatment response [4]. Identification of the underlying mechanisms contributing to this heterogeneity may aid in the search for clinically significant breast cancer subgroups [5].

There are several approaches to describing the heterogeneity of breast cancer [6,7,8,9]. Thus, the analysis of the expression of estrogen receptors, progesterone, human epidermal growth factor 2 (HER2) and the Ki-67 proliferative activity index determines the well-known molecular subtypes of breast cancer [10]. The “prognostic analysis of microarrays 50” (PAM50) is well known, which is based on the analysis of the expression of 50 genes for classification into five subtypes [11]. Based on the expression of 171 proteins identified using the reverse phase protein array (RPPA), six subtypes of breast cancer were determined [12], etc.

Determination of intermediate and final metabolic products in biological fluids and tissues provides information on ongoing cellular processes in breast cancer, since metabolic changes are one of the hallmarks of cancer [13]. Three natural metabolic clusters have been identified that differ in the levels of glycerophosphocholine and phosphocholine (1), glucose (2), and lactate and alanine (3) [5]. Another study identified subtypes depending on bile acid biosynthesis (1), the methionine pathway (2), and the metabolism of fatty acids, nucleotides, and glucose (3). The degree of metabolic changes was highly correlated with tumor aggressiveness and treatment outcome [14]. Combining transcriptomic and metabolomic data revealed three subgroups of luminal A cancer with distinct metabolic profiles [15]. Thus, understanding the metabolic changes characteristic of different subgroups of breast cancer allows us to identify new subtypes or justify the use of metabolic therapy in existing subgroups [16].

It is known that alterations in the metabolic profile of saliva are observed in breast cancer [17,18,19], which can also be used for metabolic clustering of patients. Of interest is not only the study of the metabolic profile of saliva before treatment, as was carried out in the studies described above, but also a comparison of changes in salivary metabolite levels before and after breast cancer surgery. Such studies have not previously been conducted for breast cancer, although there are data on changes in the metabolic profile of saliva before and after treatment for a number of other oncological pathologies [20,21,22]. Nevertheless, it has been shown that changes in salivary metabolites in patients after surgery can provide important information for basic research [22].

We previously analyzed the metabolic characteristics of saliva in breast cancer patients across 36 parameters and identified salivary biochemical parameters whose changes were associated with the disease [23]. In this study, we focused only on significant parameters and tested whether the previously obtained data were reproducible in a new cohort of patients. This is the first study to evaluate changes in salivary biochemical parameters four weeks after breast cancer surgery before the start of adjuvant chemotherapy.

The aim of this study was to analyze changes in the metabolic profile of saliva before and after surgical removal of a breast tumor.

## 2. Materials and Methods

### 2.1. Study Design

The study involved 660 patients with breast cancer (age 54.6 ± 1.9 years) and 127 healthy volunteers (age 49.3 ± 1.5 years). Patients were hospitalized for surgical treatment or the first course of chemotherapy. The inclusion criteria for the study were as follows: age over 18 years, no clinically significant comorbidities (diabetes mellitus, kidney disease, etc.), no other oncological diseases except breast cancer, no inflammatory diseases of the oral cavity, and histological verification of the diagnosis. The study group before treatment included patients of the following stages: stage IA + IB—228 (34.5%), stage IIA + IIB—272 (41.2%), stage III—97 (14.7%) and stage IV—63 (9.6%). Taking into account the breast cancer phenotype, the study group included Luminal A-like—234 (35.4%), Luminal B-like (HER2-negative)—200 (30.3%), Luminal B-like (HER2-positive)—60 (9.1%), HER2-enriched (Non-Lum)—44 (6.7%) and Triple-negative breast cancer (TNBC)—122 (18.5%).

Healthy volunteers were recruited from the blood transfusion department. No breast diseases were detected in volunteers in the control group, based on routine mammography or ultrasound examinations.

In 139 patients who had undergone radical mastectomy, these parameters were re-measured in saliva 4 weeks after surgery. The four-week interval is determined by the timing of the initiation of adjuvant chemotherapy. The study group after surgery included patients with the following stages: stage IA + IB—40 (28.8%), stage IIA + IIB—72 (51.8%), and stage III—27 (19.4%). Taking into account the breast cancer phenotype, the study group included Luminal A-like—27 (19.4%), Luminal B-like (HER2-negative)—74 (53.2%), Luminal B-like (HER2-positive)—10 (7.2%), HER2-enriched (Non-Lum)—6 (4.3%) and TNBC—22 (15.9%).

### 2.2. Collection, Storage, Pre-Treatment and Analysis of Saliva

Sample collection was carried out as described previously [17,23], and biochemical analysis of saliva was performed without storage or freezing. All analyses were performed in two analytical runs. Commercial control sera (cat. No. B-8213 and B-8216, Vector-Best LLC, Novosibirsk, Russia) and control materials containing the corresponding enzymes and validated methods (cat. No. B-8227, Vector-Best LLC, Novosibirsk, Russia) were used to assess accuracy and reproducibility. Enzyme activity was determined using multipoint kinetics, which is considered the most accurate method for measuring activity.

In all saliva samples, the following were determined: total protein by reaction with pyrogallol red (mg/L), urea by the urease-salicylate method according to Berthelot (mmol/L), total content of α-amino acids by reaction with ninhydrin (α-AAs, mmol/L), imidazole compounds by the diazotization reaction in the presence of sulfanilic acid (ICs, mmol/L), nitric oxide by the content of stable products of its oxidation—nitrate ions (NO, μmol/L), gamma-glutamyl transferase activity by the kinetic method using L-gamma-glutamyl-3-carboxy-4-nitroanilide as a substrate according to Seitz-Persin (GGT, U/L), lactate dehydrogenase by the kinetic UV method by the rate of nicotinamide adenine dinucleotide (NADH) oxidation (LDH, U/L), α-amylase by the kinetic method by the hydrolysis reaction CNP-oligosaccharide with the formation of 2-chloro-4-nitrophenol (U/L), catalase by determining the rate of hydrogen peroxide utilization (nkat/L). Catalase activity was determined using a commercial Servicebio kit (cat. No. G4307-48T, Wuhan Servicebio Technology, Wuhan, China). Determination of protein (cat. No. B-8084), urea (cat. No. B-8074), GGT (cat. No. B-8030), LDH (cat. No. B-8321) and α-amylase (cat. No. B-8096) was carried out using ready-made commercial kits of Vector-Best LLC (Novosibirsk, Russia) using the StatFax 3300 semi-automatic biochemical analyzer (Awareness Technology, Palm City, FL, USA).

The α-amino acid determination method is based on the fact that heating ninhydrin (trikehydrindene hydrate, C_9_H_6_O_4_, CAS 485-47-2, HiMedia Laboratories, Thane, Maharashtra, India) with substances containing primary amino groups (—NH_2_) in an alkaline medium results in the formation of a product with a stable, intense blue-violet color. A 200 μL saliva sample was diluted to 3 mL with water, and 100 μL of borate buffer (pH = 9.3, cat. No. 8500-6782, Agilent, Santa Clara, CA, USA) and 200 μL of a 0.5% aqueous ninhydrin solution were added to the sample. The samples were mixed and placed in a well-boiling water bath for at least 20 min. The samples were cooled, and their optical density was determined spectrophotometrically at 540 nm against distilled water in a 10 mm thick cuvette. Concentration was calculated using a calibration curve. A 0.002 M glycine solution (CAS 56-40-6, HiMedia Laboratories, Thane, Maharashtra, India) was used to plot the calibration graph. The absence of blue coloration in the sample indicates a low content of amino nitrogen in saliva (<0.15 nmol/L).

The method for detecting imidazole compounds is based on the diazotization of sulfanilic acid to form diazobenzenesulfonic acid. Diazobenzenesulfonic acid reacts with histidine (or tyrosine) to form an orange-red azo dye. A 200 µL sample of saliva was diluted to 3 mL with water, and then 100 µL of 0.5% sulfanilic acid (CAS 121-57-3, Central Drug House, Mumbai, Maharashtra, India), 5% sodium nitrite (CAS 7632-00-0, Central Drug House, Mumbai, Maharashtra, India), and 10% sodium carbonate (CAS 497-19-8, HiMedia Laboratories, Thane, Maharashtra, India) were added successively. The sample was thoroughly mixed, and its optical density was determined spectrophotometrically at 490 nm against a blank sample in a 10 mm thick cuvette. Concentration was calculated using a calibration curve. To plot the calibration graph, a 0.001 M histidine solution (CAS 5934-29-2, HiMedia Laboratories, Thane, Maharashtra, India) was used, which was prepared using a standard glycine solution to stabilize the diazo reaction.

The intensity of nitric oxide synthesis was determined by capillary electrophoresis (KAPEL-105M, Lumex, St. Petersburg, Russia). A deuterium lamp serves as the light source, and a diffraction monochromator with a spectral range of 190–380 nm and a spectral interval of 20 nm serves as the dispersing element, along with a photometric detector. The high-voltage DC power supply operates from 1 to 25 kV, with a 1 kV step, reversible polarity, and a current of 0–200 μA. The capillary is a quartz capillary (total length 60 cm, effective length 50 cm, internal diameter 75 μm) with liquid cooling and setting and control of the coolant temperature (range from −10 to +30 °C). The instrument operates on a 187–242 V, 50/60 Hz power supply. The saliva aliquot volume was 100 μL, preliminary precipitation of salivary proteins with 10% trichloroacetic acid solution (CAS 76-03-9, Pallav Chemicals & Solvents Pvt. Ltd., Maharashtra, India), dilution 20 times with bidistilled water, centrifugation at 7000× *g* for 5 min (CLb-16, Moscow, Russia) was carried out. Analysis conditions: pneumatic sample introduction into the capillary (30 mbar, 10 s), constant voltage of −17 kV, wavelength of the photometric detector of 374 nm, temperature of 20 °C, analysis time of 6–7 min. The accuracy of analyte determination was checked using the added-found method; the determination error does not exceed 10% over the entire concentration range. The composition of the leading electrolyte was as follows: 10 mM CrO_3_ (CAS 1333-82-0, Fluka, Buchs, Switzerland), 30 mM diethanolamine (CAS 111-42-2, Fluka, Buchs, Switzerland), 2 mM cetyltrimethylammonium hydroxide (CAS 505-86-2, Fluka, Buchs, Switzerland).

### 2.3. Statistical Analysis

The results were processed nonparametrically using Statistica 13.3 EN software (StatSoft, Tulsa, OK, USA). The results are presented as the median and interquartile range. Principal component analysis (PCA) was performed using the PCA program in R (RStudio, version 3.2.3, Boston, MA, USA). Variables for the PCA method were selected based on the results of comparing biochemical parameters in the study groups. When comparing two groups, we used the Mann–Whitney test; when comparing three or more groups, we used the Kruskal–Wallis test. Next, we selected the parameters for which differences between all groups were significant at the *p* < 0.05 level. The parameters were standardized by reducing the data to zero mean and unit variance. The PCA results are presented as factor planes and corresponding correlation circles. In each case, only the first two principal components (PC1 and PC2) are shown in the figures. The color of the arrows on the correlation circle changes from blue (weak correlation) to red (strong correlation), as shown in the color bar. The orientation of the arrows characterizes positive and negative correlations (for the first principal component, we analyze the arrows’ position relative to the vertical axis; for the second principal component, we analyze the arrows’ position relative to the horizontal axis).

## 3. Results

### 3.1. Changes in the Biochemical Composition of Saliva After Breast Cancer Surgery

In breast cancer, the activity of GGT (+17.6%, *p* < 0.0001), catalase (+14.7%, *p* = 0.0485), urea content (+24.6%, *p* = 0.0006), total protein (+13.6%, *p* = 0.0432), α-amino acids (+3.1%, *p* < 0.0001) increased in saliva, but LDH activity (−16.9%, *p* < 0.0001) and the content of imidazole compounds (−25.2%, *p* < 0.0001) decreased (Table 1).

At 4 weeks after surgery, the content of total protein (−25.0%, *p* = 0.0001), urea (−2.2%), α-amino acids (−1.5%), imidazole compounds (−37.4%, *p* = 0.0274), as well as the activity of GGT (−5.1%, *p* = 0.0017), LDH (−12.4%) and α-amylase (−11.3%) decreased compared to the values before surgery (Figure 1). Only for the content of NO and salivary catalase activity a slight increase was observed in the postoperative period (Figure 1).

The PCA method showed that the salivary metabolic profile of the control and postoperative groups was similar and significantly different from that of the primary breast cancer group (*p* = 0.0026) (Figure 2). Only parameters whose differences from the healthy controls were significant at a *p* < 0.05 level were selected for PCA (Table 1).

**Table 1 metabolites-15-00693-t001:** Biochemical composition of saliva before treatment.

Indicators	Breast Cancer, n = 660	Healthy Controls, n = 127	*p*-Value
Protein, g/L	0.75 [0.51; 1.00]	0.66 [0.48; 0.97]	0.0432
Urea, mmol/L	9.31 [6.27; 12.23]	7.47 [5.30; 10.58]	0.0006
α-AAs, mmol/L	3.97 [3.74; 4.34]	3.85 [3.70; 4.07]	<0.0001
Catalase, ncat/mL	3.43 [2.51; 5.15]	2.99 [2.47; 4.03]	0.0485
LDH, U/L	1062.0 [628.0; 1623.0]	1277.0 [833.7; 1563.0]	<0.0001
GGT, U/L	22.7 [19.7; 25.9]	19.3 [17.0; 22.7]	<0.0001
α-Amylase, U/L	256.2 [149.1; 535.1]	199.6 [103.4; 421.8]	0.5378
ICs, mmol/L	0.246 [0.131; 0.405]	0.329 [0.211; 0.531]	<0.0001
NO, μmol/L	75.25 [67.00; 84.83]	73.61 [69.72; 79.35]	0.8070

Note. Here, and further in Table 2 and Table 3, the data are presented as median and interquartile range Me [25%Q; 75%Q].

**Table 2 metabolites-15-00693-t002:** Biochemical composition of saliva before and after surgery depending on the stage of breast cancer.

Indicators	Surgery	St I, n = 228/40	St II, n = 272/72	St III, n = 97/27	St IV, n = 63/0
Protein, g/L	BS	0.78 [0.59; 1.06]	0.77 [0.53; 1.05]	0.66 [0.39; 0.81]	0.72 [0.47; 0.93]
		*p* = 0.0087 *	*p* = 0.0336	-	-
	AS	0.64 [0.48; 0.81]	0.55 [0.38; 0.77]	0.72 [0.50; 0.93]	-
		-	*p* = 0.0330	-	-
Urea, mmol/L	BS	10.19 [7.41; 12.73]	9.57 [6.61; 12.22]	7.59 [4.50; 10.50]	7.32 [4.08; 11.68]
		*p* = 0.0000	*p* = 0.0001	-	-
	AS	9.43 [6.99; 12.28]	7.74 [5.38; 11.67]	10.53 [6.51; 12.72]	-
		*p* = 0.0050	-	*p* = 0.0080	-
α-AAs, mmol/L	BS	4.02 [3.76; 4.40]	4.01 [3.78; 4.42]	3.88 [3.67; 4.16]	3.87 [3.69; 4.32]
		*p* = 0.0001	*p* = 0.0000	-	-
	AS	4.08 [3.80; 4.45]	3.87 [3.71; 4.16]	3.95 [3.74; 4.28]	-
		*p* = 0.0032	-	-	-
Catalase, ncat/mL	BS	3.35 [2.40; 5.05]	3.61 [2.61; 5.05]	3.34 [2.34; 5.51]	3.09 [2.44; 4.24]
		-	*p* = 0.0150	-	-
	AS	3.93 [2.83; 5.13]	3.33 [2.57; 4.40]	3.46 [2.20; 4.90]	-
LDH, U/L	BS	1049.5 [571.4; 1605.5]	1106.0 [674.8; 1713.0]	1023.0 [661.2; 1503.0]	930.4 [542.0; 1452.0]
		-	-	-	*p* = 0.0178
	AS	992.2 [753.6; 1657.0]	927.9 [513.6; 1393.0]	784.3 [627.4; 1474.0]	-
		-	*p* = 0.0039	-	-
GGT, U/L	BS	23.4 [20.9; 26.7]	22.7 [19.6; 26.1]	22.0 [18.3; 24.7]	22.4 [18.5; 24.5]
		*p* = 0.0000	*p* = 0.0000	*p* = 0.0048	*p* = 0.0059
	AS	23.0 [19.0; 25.1]	21.4 [17.8; 24.1]	20.3 [18.8; 23.3]	-
		*p* = 0.0012	*p* = 0.0183	-	-
ICs, mmol/L	BS	0.234 [0.136; 0.388]	0.234 [0.123; 0.413]	0.289 [0.145; 0.436]	0.252 [0.159; 0.357]
		*p* = 0.0000	*p* = 0.0003	-	*p* = 0.0121
	AS	0.172 [0.087; 0.361]	0.195 [0.118; 0.392]	0.153 [0.080; 0.285]	-
		*p* = 0.0028	*p* = 0.0007	*p* = 0.0002	-

Note. * *p*-value is shown compared to healthy control. BS—before surgery, AS—after surgery.

**Table 3 metabolites-15-00693-t003:** Biochemical composition of saliva before and after surgery depending on the molecular biological subtype of breast cancer.

Indicators	Sur	Lum A, n = 234/27	Lum B(−), n = 200/74	Lum B(+), n = 60/10	Non-Lum, n = 42/6	TNBC, n = 122/22
Protein, g/L	BS	0.84 [0.59; 1.11]	0.78 [0.59; 1.02]	0.75 [0.50; 0.88]	0.63 [0.45; 0.78]	0.63 [0.42; 0.86]
		*p* = 0.0007	*p* = 0.0193	-	-	-
	AS	0.64 [0.47; 0.93]	0.59 [0.38; 0.85]	0.68 [0.43; 0.88]	0.51 [0.39; 0.68]	0.61 [0.46; 0.78]
Urea, mmol/L	BS	10.46 [7.69; 12.62]	9.52 [6.72; 12.52]	8.26 [5.27; 12.07]	7.20 [4.72; 9.86]	6.87 [4.08; 10.48]
		*p* = 0.0000	*p* = 0.0002	-	-	-
	AS	11.24 [7.19; 12.67]	8.59 [5.98; 11.19]	10.44 [5.32; 13.99]	11.55 [9.77; 13.59]	7.49 [5.53; 12.35]
		*p* = 0.0008	-	-	*p* = 0.0453	-
α-AAs, mmol/L	BS	4.05 [3.79; 4.39]	4.00 [3.74; 4.41]	3.96 [3.72; 4.24]	3.81 [3.62; 4.09]	3.88 [3.69; 4.19]
		*p* = 0.0000	*p* = 0.0003	-	-	-
	AS	4.06 [3.68; 4.32]	3.88 [3.71; 4.22]	4.11 [3.94; 4.61]	3.96 [3.74; 4.22]	3.90 [3.78; 4.33]
		-	-	*p* = 0.0096	-	-
Catalase, ncat/mL	BS	3.47 [2.57; 5.30]	3.39 [2.37; 5.07]	3.98 [2.43; 5.95]	3.09 [2.45; 4.79]	3.49 [2.59; 4.90]
		*p* = 0.0242	-	*p* = 0.0416	-	*p* = 0.0375
	AS	3.46 [2.23; 4.51]	3.58 [2.67; 4.80]	3.03 [2.43; 3.52]	3.75 [2.00; 6.73]	3.29 [2.74; 5.06]
LDH, U/L	BS	1203.0 [686.2; 1727.0]	1020.0 [598.1; 1652.0]	1050.0 [674.8; 1353.0]	1009.5 [710.1; 1513.0]	931.7 [522.8; 1431.5]
		-	-	-	-	*p* = 0.0025
	AS	1228.0 [747.4; 1629.0]	950.8 [578.5; 1380.0]	924.9 [539.3; 1219.0]	842.1 [757.0; 968.4]	909.0 [709.6; 1717.0]
GGT, U/L	BS	23.1 [20.0; 26.8]	22.8 [19.6; 25.9]	22.9 [20.0; 26.2]	21.7 [18.4; 23.8]	22.6 [19.3; 24.9]
		*p* = 0.0000	*p* = 0.0000	*p* = 0.0001	*p* = 0.0254	*p* = 0.0000
	AS	21.8 [18.3; 25.9]	21.5 [18.8; 23.6]	22.6 [21.2; 26.6]	20.6 [18.8; 22.5]	19.4 [17.2; 24.3]
		*p* = 0.0441	*p* = 0.0024	*p* = 0.0133	-	-
ICs, mmol/L	BS	0.237 [0.131; 0.396]	0.219 [0.104; 0.376]	0.307 [0.207; 0.418]	0.279 [0.160; 0.362]	0.272 [0.153; 0.459]
		*p* = 0.0001	*p* = 0.0000	-	*p* = 0.0476	*p* = 0.0344
	AS	0.178 [0.083; 0.393]	0.167 [0.101; 0.319]	0.171 [0.071; 0.219]	0.218 [0.052; 0.320]	0.415 [0.171; 0.601]
		-	-	*p* = 0.0150	-	-

Note. BS—before surgery, AS—after surgery.

According to the first principal component, the maximum contribution to the separation was made by LDH activity (*r* = 0.6892), protein concentrations (*r* = 0.6874), urea (*r* = 0.6653), catalase activity (*r* = 0.5581) and GGT (*r* = 0.4936). The separation by the first principal component was statistically significant (*p* = 0.00264). According to the second principal component, the maximum contribution to the separation was made by α-AMA (*r* = 0.7323), ICs (*r* = 0.4320), GGT activity (*r* = −0.4113) and catalase (*r* = −0.4156). The separation by the second principal component was also statistically significant (*p* = 0.00451).

### 3.2. The Influence of Breast Cancer Stage on Changes in Salivary Biochemical Parameters Before and After Surgery

Before surgery, a decrease in the concentration of total protein, urea, α-amino acids, and the activity of catalase, GGT, and LDH in saliva was observed with increasing breast cancer stage (Table 2).

After surgery, a decrease in most parameters was observed, with changes in advanced stages of breast cancer being more pronounced than in early stages (Figure 3). Thus, for protein and urea, the concentration increased after surgery at stage III (+8.5 and +38.8%, respectively), while for LDH (−23.3%), GGT (−7.7%), and imidazole compounds (−47.1%), the maximum decrease relative to preoperative values was observed at stage III.

PCA showed that, before breast cancer surgery, stages I and II were similar in salivary biochemical profiles and significantly differed from both healthy controls and advanced stages (Figure 4A). After breast cancer surgery, no subgroup separation was observed in the factorial diagram (*p* = 0.1091) (Figure 4B).

Before breast cancer surgery, the maximum contribution to the separation of subgroups according to the first principal component was made by protein (*r* = 0.6819), LDH (*r* = 0.6660), and urea (*r* = 0.6536), according to the second principal component—α-amino acids (*r* = 0.7729). The separation of subgroups according to both principal components was statistically significant (*p* = 0.0001 and *p* = 0.0045, respectively) (Figure 4A).

### 3.3. The Influence of Molecular Biological Subtype of Breast Cancer on Changes in Biochemical Parameters of Saliva Before and After Surgery

It was shown that the values of most biochemical parameters in saliva decreased after surgery regardless of breast cancer phenotype (Table 3, Figure 5). The exceptions were urea, α-amino acids, and catalase (Figure 5).

The PCA method showed that luminal A and luminal B HER2-negative breast cancer subtypes differed the most from the healthy control in the biochemical profile of saliva (Figure 6A).

The remaining subtypes were not separated in the factor diagram either from each other or from the healthy control (Figure 6A). Before breast cancer surgery, the first principal component allowed us to distinguish luminal A and luminal B HER2-negative subtypes by the contribution of protein (*r* = 0.6820), LDH (*r* = 0.6690), urea (*r* = 0.6689) and catalase (*r* = 0.5498), the second principal component—by the content of α-amino acids (*r* = 0.7516). The separation of subgroups by the first principal component is statistically significant (*p* < 0.0001) (Figure 6A). After surgery, the greatest differences were shown for TNBC only in the second principal component for the contribution of imidazole compounds (*r* = 0.7516) and α-amino acids (*r* = 0.4839) (*p* = 0.0015) (Figure 6B).

## 4. Discussion

The relationship between breast cancer and salivary metabolites is controversial. An increase in urea concentration in saliva has been observed, which is most pronounced for luminal subtypes and in the early stages of breast cancer. It is assumed that the concentration of urea in saliva is determined by the composition of the oral microbiome [24]. We hypothesized that since the oral microbiome is most diverse in estrogen-receptor-positive cancer and least diverse in TNBC, the increased urea concentration in luminal breast cancer may be the result of hormone-dependent changes in the microbiome in this group of patients, which, however, requires further verification [25]. It was precisely with initially low urea concentrations that an increase in its concentration was observed after tumor removal, which may be indirectly related to changes in the composition of the oral microbiome after treatment.

Salivary protein concentrations decreased after breast cancer surgery for all phenotypes, but changes were less pronounced for TNBC. Salivary α-amino acid content increased most significantly in the HER2-positive breast cancer subgroups. We have previously shown that HER2-positive breast cancer is characterized by an increase in free amino acid levels compared to healthy controls and other molecular biological subtypes of breast cancer [26]. Regarding imidazole compounds, an increase in their concentration was observed only in TNBC after surgery. Imidazole derivatives include the amino acid histidine and its metabolites (histamine, urocanilic acid, etc.). Histamine is a potent angiogenic factor and has an immunosuppressive effect, and therefore a decrease in histamine concentrations after breast cancer surgery may reflect an increase in the immune response, which is not observed in TNBC. The study demonstrated the presence of histamine receptors (H3R and H4R) in human breast tissue, and therefore H3R may be involved in the regulation of breast cancer growth and progression, representing a new molecular target for a therapeutic approach [27,28].

GGT is known to play a key role in the homeostasis of glutathione (GSH), the main component of the antioxidant defense [29]. GGT expression is increased in breast cancer, and GGT activity affects the rate of cell growth and increases tumor resistance to chemotherapy [30,31]. We have shown an increase in the activity of antioxidant defense enzymes (catalase, GGT), as well as the level of NO in saliva in breast cancer. This is consistent with the known fact of intensification of lipid peroxidation in breast cancer, a pronounced inflammatory response and a compensatory increase in the efficiency of the antioxidant defense system [32]. In the study, the authors showed that increased lipid peroxidation in blood plasma were accompanied by increased glutathione peroxidase activity, which confirms other data on changes in antioxidant homeostasis in breast cancer [33]. After surgery, enzyme activity changed ambiguously; no specific relationships were shown for salivary catalase, while GGT activity decreased, which was especially pronounced at advanced stages and in TNBC.

Salivary LDH activity reflects the condition of the oral mucosal epithelium [34]. This enzyme is normally found in the cytoplasm of most cells in the human body, but is released outside the cell under oxidative stress [35]. Therefore, assessing salivary LDH levels can serve as an effective tool for assessing diseases that compromise the integrity of the oral mucosa [36]. It has also been shown that the salivary LDH isoenzymes profile has a completely different structure than that in plasma, similar to the structure found in the oral epithelium, indicating that the main source of LDH in saliva is likely oral epithelial cells [37]. In general, we have shown a decrease in salivary LDH activity in breast cancer, with LDH activity being higher in the most prognostically favorable luminal A subtype, while the minimum salivary LDH activity value corresponded to TNBC. We have previously shown that it is in luminal A breast cancer, as well as in the early stages, that the levels of inflammatory salivary cytokines are increased [38]. It can be assumed that the relative increase in LDH activity in the luminal A phenotype is associated with the presence of systemic inflammation in this subtype of breast cancer. We also observe a decrease in LDH activity after breast cancer surgery, and the decrease in activity is more pronounced for advanced stages and the HER2-positive phenotype of breast cancer. The important role of oral microflora in the process of salivary LDH utilization should not be excluded, which requires separate detailed consideration. Thus, Feng et al. found a relative amount of *Bacteroidetes* and *Firmicutes* in saliva samples of both luminal A and luminal B breast cancer, whereas HER2 and TNBC types demonstrated higher levels of *Proteobacteria* [39]. *Streptococcus* was present in large numbers in all pathological types, but slightly less in TNBC; *Neisseria* was most common in luminal A and least common in TNBC. Conversely, the HER2 type demonstrated a higher abundance of *Porphyromonas*. Saliva samples showed associations of the luminal B subtype of breast cancer with both *Proteobacteria* and *Bacteroidetes*, complemented by a clear association between TNBC and *Actinobacteria*. This suggests that the microbial matrix of saliva may provide specific information about specific breast cancer subtypes.

An increase in salivary α-amylase activity is considered a marker of stress in the body, along with the hormone cortisol, because of activation of the sympathoadrenergic system [40]. Another explanation for the increase in α-amylase activity may be its interaction with growth-stimulating hormones, such as estrogens [41]. With both interpretations, a decrease in salivary α-amylase activity after tumor removal is justified.

Overall, it has been shown that several salivary biochemical parameters are restored after breast cancer surgery; however, these values do not reach those of healthy controls. A 4-week period is apparently insufficient for complete restoration of the salivary metabolic profile. Before surgery, a statistically significant separation of the subgroups with breast cancer and healthy controls was observed in the PCA factorial diagram. Differences in the salivary metabolic profile were also pronounced in early-stage breast cancer and in the prognostically favorable luminal A and B HER2-negative subtypes. After surgery, the salivary metabolic profile did not differ statistically significantly from that of healthy controls and was independent of breast cancer stage. Differences were demonstrated only for TNBC compared to other breast cancer subtypes after surgery, which may be due to the more aggressive nature of this breast cancer subtype.

Understanding how salivary biochemical parameters change before and after breast cancer surgery may be useful for identifying salivary biomarkers. In this case, a decrease in marker concentration after surgery may indicate a correlation with breast cancer. Insufficient recovery of salivary biochemical parameters after surgery may be due to the presence of residual tumor or metastatic lesions, as well as a sign of progression, requiring careful attention and additional diagnostics. Timely correction of excessive oxidative stress deserves special attention, as this may improve the patient’s recovery. However, the described clinical application potential is currently debatable and requires clinical testing.

Limitations of the study include the fact that HER2-positive status is not an indication for breast cancer surgery, making the proportion of these patients in the postoperative sample very low. The study is also limited by the fact that only a four-week period was used to assess changes in salivary biochemical parameters. Since salivary biochemical parameters may be distorted after the initiation of chemotherapy, extending the period for this category of patients is not feasible. Therefore, we limited ourselves to two points: before surgery and before the initiation of adjuvant chemotherapy. Patients who are not prescribed adjuvant chemotherapy undergo radiation therapy, which also does not allow for an accurate assessment of metabolic changes. We also did not assess the oral microbiome, although we have hypothesized a number of hypotheses regarding the influence of microflora on enzyme activity and substrate levels in saliva. We plan to address this issue in future studies. It should be noted that the range of parameters measured in this study is limited, but we also plan to expand it to provide a more complete picture of changes in the salivary metabolic profile before and after breast cancer surgery.

## 5. Conclusions

Following surgical treatment for breast cancer, a number of salivary biochemical parameters are restored. A four-week interval is insufficient for complete recovery; however, for most biochemical parameters, the values are intermediate between those in the control group and the group of patients with primary breast cancer. This confirms that the observed change in the salivary metabolic profile is due to the presence of breast cancer in these patients. It was shown that the greatest deviations in the salivary metabolic profile from the norm before surgery corresponded to early stages and the most favorable prognostic phenotypes of breast cancer. Postoperatively, however, the most pronounced metabolic differences were observed for TNBC.

## Figures and Tables

**Figure 1 metabolites-15-00693-f001:**
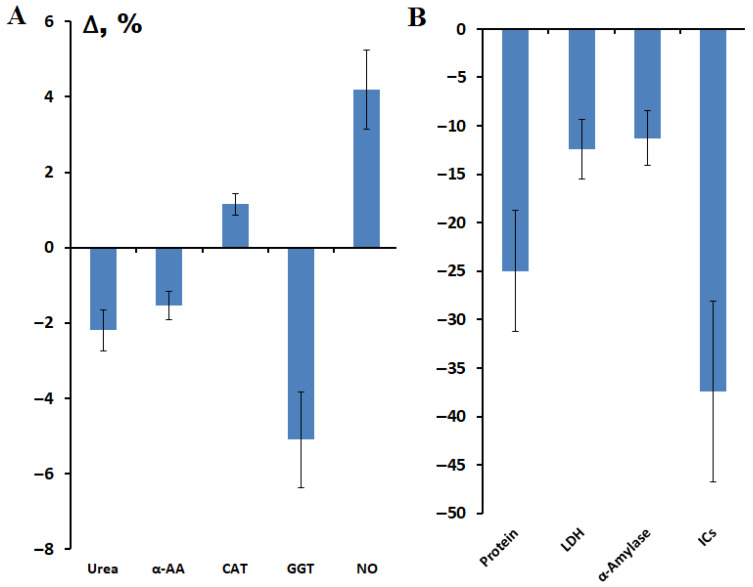
Changes in the biochemical composition of saliva after breast cancer surgery, %: (**A**)—urea, α-AA, CAT, GGT and NO; (**B**)—protein, LDH, α-amylase and ICs. α-AA—α-amino acids, CAT—catalase, GGT—gamma-glutamyl transferase, NO—nitric oxide, LDH—lactate dehydrogenase, ICs—imidazole compounds. Here and below, Δ is the relative concentration, which is calculated as the difference between the concentration value before and after surgery, divided by the concentration value before surgery as a percentage.

**Figure 2 metabolites-15-00693-f002:**
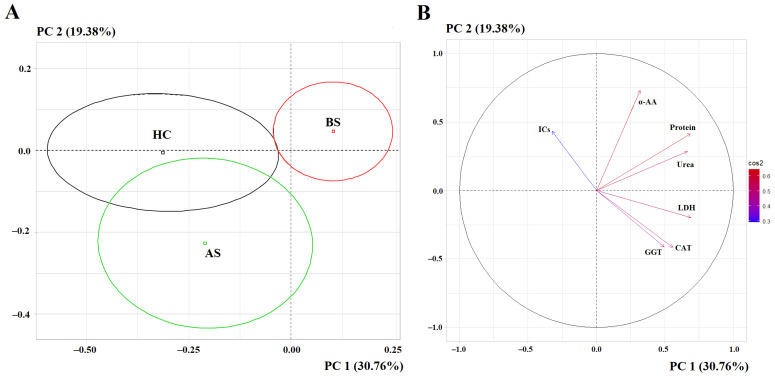
PCA factorial diagram (**A**) and analysis of variables (**B**) for comparison of three groups: healthy control (HC), breast cancer before surgery (BS), breast cancer after surgery (AS). α-AA—α-amino acids, CAT—catalase, GGT—gamma-glutamyl transferase, LDH—lactate dehydrogenase, ICs—imidazole compounds, PC—principal component. The color of the arrows on the correlation circle changes from blue (weak correlation) to red (strong correlation), as shown in the color bar.

**Figure 3 metabolites-15-00693-f003:**
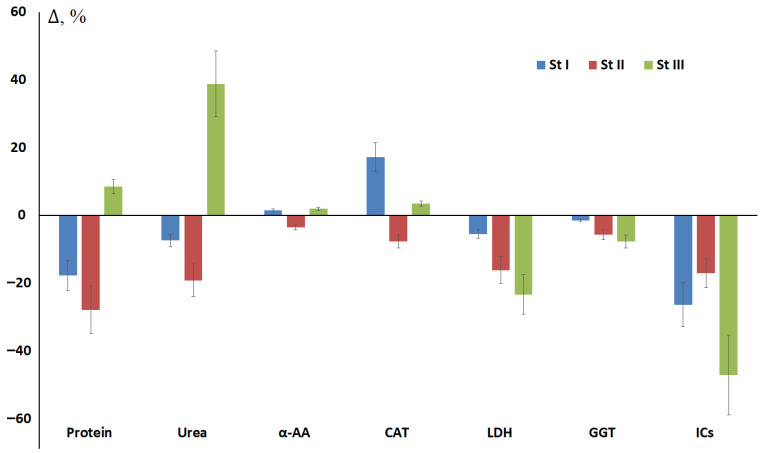
Relative values of biochemical composition of saliva after surgery compared with values before breast cancer surgery depending on the stage. α-AA—α-amino acids, CAT—catalase, GGT—gamma-glutamyl transferase, LDH—lactate dehydrogenase, ICs—imidazole compounds.

**Figure 4 metabolites-15-00693-f004:**
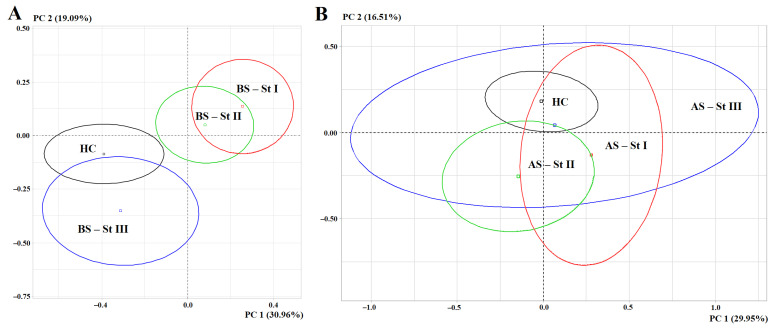
Individuals factor map (PCA): (**A**)—before surgery (*p* = 6.074026 × 10^−6^); (**B**)—after surgery (*p* = 0.1091366). HC—healthy controls, BS—before surgery, AS—after surgery, PC—principal component.

**Figure 5 metabolites-15-00693-f005:**
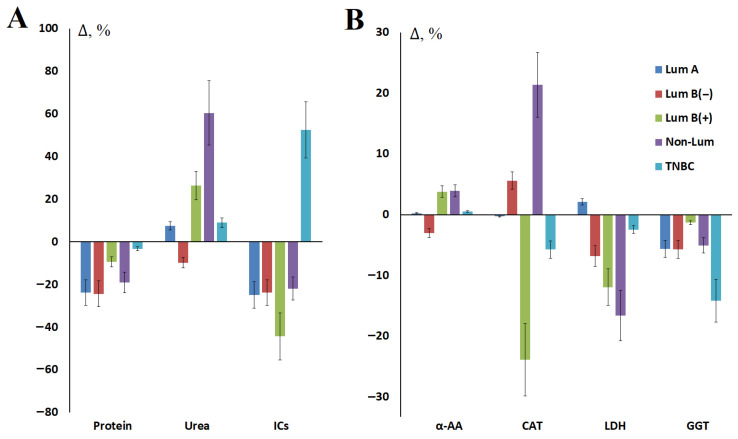
Relative indices of salivary biochemical composition after surgery compared with values before breast cancer surgery depending on the molecular biological subtype of breast cancer: (**A**)—Protein, urea and ICs; (**B**)—α-AAs, catalase, LDH, and GGT.

**Figure 6 metabolites-15-00693-f006:**
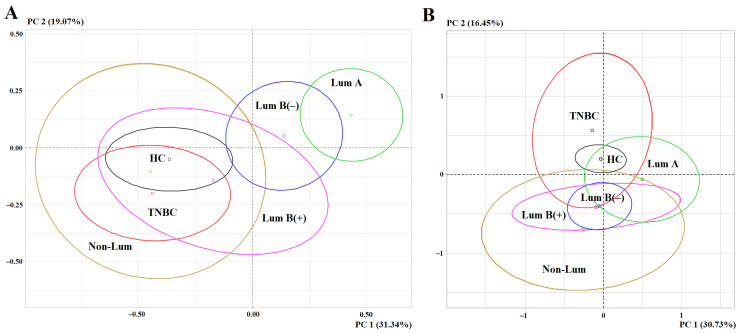
Individuals factor map (PCA): (**A**)—before surgery (*p* = 2.265219 × 10^−8^); (**B**)—after surgery (*p* = 0.001461923). HC—healthy controls, PC—principal component, Lum A—luminal A, Lum B (0)—luminal B HER2-negative, Lum B (+)—luminal B HER2-positive, Non-Lum—HER2-enriched and TNBC—triple-negative breast cancer.

## Data Availability

The raw data supporting the conclusions of this article will be made available by the authors on request.

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
