# Peer review of "Metabolic Features of Saliva Before and After Breast Cancer Surgery"

_metabolites, 2025, doi:10.3390/metabo15110693_

Round 1

Reviewer 1 Report

Comments and Suggestions for Authors

The research manuscript entitled “Metabolic Features of Saliva before and after Breast Cancer Surgery” reports pre- and post-surgery changes in salivary biochemical markers in 660 breast cancer patients and 127 healthy controls, and follow-up on 139 patients four weeks post-surgery. It measured some salivary metabolites and enzyme activities, and the results were submitted to nonparametric tests and principal component analysis (PCA) for determining association with cancer stage and molecular subtype. The authors conclude that a number of salivary metabolites partially recover after surgery, with diagnostic and prognostic implications for salivary metabolomics in the clinical management of breast cancer. Generally, the manuscript deals with an interesting and relevant subject and is within the scope of Metabolites, i.e., metabolomic profiling, biochemical biomarker discovery, and metabolic changes linked to diseases. The experimental design involves a large cohort and clinically relevant endpoints. However, before the research is publishable, the manuscript requires several revisions in terms of its scientific depth, data interpretation, and methodological rigor to meet the standards of Metabolites.

  1. The study’s novelty is modest, as similar salivary metabolite analyses in breast cancer have been published previously by the same group and others (Bel’skaya, L.V.; Sarf, E.A.; Solomatin, D.V.; Kosenok, V.K. Metabolic Features of Saliva in Breast Cancer Patients. Metabolites 2022, 12, 166. https://doi.org/10.3390/metabo12020166). The manuscript should better clarify what new metabolic insights are gained post-surgery that were not known before. A clearer hypothesis statement and mechanistic discussion are needed.
  2. The use of PCA is appropriate, but no validation is reported. Please add references for enzymatic assay validation and reproducibility.
  3. The authors should clarify why they are using a less strict threshold of p<0.10 rather than p<0.05, as normal (increases the chance of finding a significant result, but it also increases the risk of a false positive compared to a stricter level) (lines 120–121).
  4. Result (section 3.1), please check and confirm the value of GGT activity change (the activity of GGT (+19.7%, p<0.0001); the value in Table 1 does not match the percentage of change shown in text.
  5. The discussion links some biochemical changes to the oral microbiome and oxidative stress, but these claims are speculative without direct microbiome data. The discussion should clearly distinguish between observed biochemical changes and hypothetical mechanisms. The discussion of urea and microbiome associations should be condensed or supported with literature data.
  6. The 4-week post-surgery time point may not adequately reflect metabolic stabilization. Clinical rationale for this time point should be provided by the authors or include previous references.
  7. The translational potential of these salivary biomarkers for either early detection or post-surgical monitoring is uncertain. A discussion section on clinical implications and potential future applications would strengthen the manuscript. Please consider adding a short paragraph to discuss this issue.
  8. The ethics approval and informed consent are appropriately mentioned, but details on sample handling (time between collection and analysis, quality control measures) should be added for reproducibility. Line 95–96, the authors described that “biochemical analysis of saliva was performed without storage or freezing”; however, the reviewer wondered that with the large quantity of samples, how the authors can handle and analyze samples right away after collection. In the Methods, clarify whether fasting samples were collected and at what time of day, as salivary metabolites are diurnally variable.
  9. Abbreviations (e.g., TNBC) need to be standardized and defined on initial use in text and abstract.
  10. Improve figure quality (Figures 2, 4, and 6) and ensure all axes and symbols are clearly labeled.

Reviewer 2 Report

Comments and Suggestions for Authors

Summary

The authors collected a large saliva sample set from patients and healthy volunteers. They analyzed the differences between healthy controls and breast cancer patients, as well as the changes in breast cancer patients before or after surgery, in several biochemical indicators, such as proteins, amino acids, urea, and others. They found that the biochemical parameters of the patients after surgery are between those of the healthy control and primary breast cancer patients. Overall, the study provides insight into the correlation between saliva and breast cancer surgery and will serve as a valuable dataset for the scientific community.

Major point

  1. The authors should provide a tidy Excel data sheet containing all relevant sample information, including: (1) all of the biochemical parameters, (2) age, (3) sex, (4) weight, (5) Height, (6) cancer type, (7) stage of the breast cancer, and (8) before or after surgery.
  2. As the study involves saliva samples, the authors should clarify the sample collection conditions. Specifically, the authors should indicate whether samples were collected before or after meals, or at random times and also reveal how the samples were preserved before the analysis.
  3. The authors should provide detailed methodological information regarding the determination of each biochemical parameter, such as the method to measure the amino acid reaction, imidazole compounds, gamma-glutamyl transferase activity, alpha amylase activity, etc.
  4. The sources of all chemicals and reagents used in the study should be specified in the paper.
  5. The statistical analysis methods applied to each dataset should be clearly described below each corresponding table.
  6. The authors should specify whether the error values presented in the tables represent standard deviation (SD) or standard error of the mean (SEM).

Minor point

The A and B label of each figure should be on the upper left site of each figure.

Round 2

Reviewer 1 Report

Comments and Suggestions for Authors

The authors have carefully addressed all previous comments and significantly improved the quality of the manuscript. The scientific content is now clear, well-structured, and consistent with the scope of Metabolites. 

Author Response

We once again express our gratitude to the reviewer for his attention to the manuscript and valuable comments.

Reviewer 2 Report

Comments and Suggestions for Authors

Considering the data is going to be published in the journal, we should include the metadata in the supporting information though the data will be part of the dissertation. However, I will leave the final decision to the editors. I have no further comments.

Author Response

We once again express our gratitude to the reviewer for his attention to the manuscript and valuable comments. We will be happy to provide the data upon request to the corresponding author; however, we do not believe it is appropriate to publish them in open access supplementary materials.